# An Efficient Time-Domain End-to-End Single-Channel Bird Sound Separation Network

**DOI:** 10.3390/ani12223117

**Published:** 2022-11-11

**Authors:** Chengyun Zhang, Yonghuan Chen, Zezhou Hao, Xinghui Gao

**Affiliations:** 1School of Electronics and Communication Engineering, Guangzhou University, Guangzhou 510006, China; 2Research Institute of Tropical Forestry, Chinese Academy of Forestry, Guangzhou 510520, China

**Keywords:** bird sound separation, transformer, deep learning, lower computational resources, dual-path network

## Abstract

**Simple Summary:**

Automatic bird sound recognition using artificial intelligence technology has been widely used to identify bird species recently. However, the bird sounds recorded in the wild are usually mixed sounds, which can affect the accuracy of identification. In this paper, we utilized massive amounts of data of bird sounds and proposed an efficient time-domain single-channel bird sound separation network. Our proposed network achieved good separation performance and fast separation speed while greatly reducing the consumption of computational resources. Our work may help to discriminate individual birds and study the interaction between individual birds, as well as to realize the automatic identification of bird species in various mobile and edge computing devices.

**Abstract:**

Bird sounds have obvious characteristics per species, and they are an important way for birds to communicate and transmit information. However, the recorded bird sounds in the field are usually mixed, which making it challenging to identify different bird species and to perform associated tasks. In this study, based on the supervised learning framework, we propose a bird sound separation network, a dual-path tiny transformer network, to directly perform end-to-end mixed species bird sound separation in the time-domain. This separation network is mainly composed of the dual-path network and the simplified transformer structure, which greatly reduces the computational resources required of the network. Experimental results show that our proposed separation network has good separation performance (SI-SNRi reaches 19.3 dB and SDRi reaches 20.1 dB), but compared with DPRNN and DPTNet, its parameters and floating point operations are greatly reduced, which means a higher separation efficiency and faster separation speed. The good separation performance and high separation efficiency indicate that our proposed separation network is valuable for distinguishing individual birds and studying the interaction between individual birds, as well as for realizing the automatic identification of bird species on a variety of mobile devices or edge computing devices.

## 1. Introduction

In recent years, acoustic monitoring has been widely used in bird monitoring, research, and protection [1]. This method does not invade or damage the natural environment, and can reduce the impact of human disturbance on birds. Audio files collected in acoustic monitoring can be used as important data to track the changes in bird community distribution with time. With the recorded audio data of birds, we can use deep learning methods to quickly identify bird species [2,3,4], understand species composition, and use acoustic indices to analyze species richness in the current environment [5,6,7]. However, the field recording environment is very complex, and the bird sounds recorded are often affected by various factors, such as noise interference, the recorded bird sound is too small, or the bird sound is mixed, etc. Mixed bird sounds are a common problem when recording bird sounds in the field [8,9], because birds are social animals and usually chirp together, and this has a direct effect on the accuracy of bird species identification [9] and species richness estimation [10,11,12]. At the same time, since distinguishing individual bird sounds is the premise of many studies, the separation of mixed species bird sounds is of great significance for bird population statistics, individual bird discrimination, and individual bird interaction studies. However, there are few research reports on bird sound separation.

Due to the rapid development of deep learning, significant progress has been made recently in speech separation [13,14]. In general, time-domain single-channel speech separation can be described with an “encoder-separator-decoder” framework, as shown in Figure 1. The encoder transforms the signal into a representation of the latent space. The separator learns the mask of different source sounds and multiplies it with the mixed sound to separate the mixed sound. The decoder inverse transforms the separated signal into a time-domain signal. TasNet [15] used the “encoder-separator-decoder” framework to model the signal directly in the time-domain and perform source separation on the output of the non-negative encoder. There was no frequency decomposition step in this method, the separation problem was simplified to estimate the source mask on the output of the encoder, and then the separated signal was synthesized using the decoder, which reduced the computational cost of speech separation and the output delay. Conv-TasNet [16] used TCN [17] instead of LSTM [18] for feature extraction and separation based on TasNet. Compared with the stepwise execution of LSTM, TCN can process in parallel while using deep separable convolution [19], which reduced the number of parameters and computing resources, thus achieving better separation performance (SI-SNRi reaches 15.3 dB and SDRi reaches 15.6 dB in the WSJ0-2mix dataset [20]). By dividing the input long feature sequence into multiple small chunks and then by using LSTM to perform local modeling within each chunk and global modeling between different chunks, dual-path recurrent neural network (DPRNN [21]) provided a good solution for long speech sequences modeling. This dual-path processing greatly improved the model’s ability to model long speech sequences, and further improved the separation performance (SI-SNRi reaches 18.8 dB and SDRi reaches 19.0 dB in the WSJ0-2mix dataset) when facing longer speech sequences, due to the advantages of transformers in learning the long-term dependence of the context-aware [22], Dual-path transformer network (DPTNet [23]) that uses transformer instead of LSTM in DPRNN, making the network more prominent in sequence modeling (SI-SNRi reaches 20.2 dB and SDRi reaches 20.6 dB in the WSJ0-2mix dataset). Besides DPTNet, Sandglasset [24], SepFormer [25], SFSRNet [26], and other networks also used transformer to conduct speech separation.

With the rapid development of time-domain single-channel speech separation, some problems arise, such as the higher and higher requirements of computing resources. Training a separation network can take as little as a few days, up to a week or two, and requires multiple high-end GPUs to compute in parallel. The huge time cost and high equipment requirements limit the application and promotion of the separation networks in various embedded systems or edge computing devices. Therefore, it is necessary to find some separation networks that are more efficient and require fewer computing resources. SuDoRM-RF [27] used multiple depthwise separable convolution modules to continuously down-sample and resample high-dimensional features to form a U-ConvBlock for the feature extraction of sequences. Since the feature length became shorter after down-sampling, the model effectively reduced the computation and memory footprint. The Global Attention Local Recurrent (GALR) network with dual-path structure [28] used RNN for local modeling, and used a self-attention mechanism for global modeling between chunks. By working alternately with these two structures, the use of the self-attention mechanism was reduced, and thus the required resources were reduced. SepFormer [25] used a two-path transformer for intra-chunk local modeling and inter-chunk global modeling for long sequences to ensure excellent performance, and a higher down-sampling dimension on the encoder to shorten the output feature length to reduce floating point operations and memory requirements. As we all know, the higher the sampling frequency of the audio, the more sampling points, and the more computational resources are consumed during separating. The paper [29] studied the application of training with a low sampling frequency, and then tested the audio with a high sampling frequency, which reduced the resource consumption during training.

It can be seen from the above that time-domain single-channel speech separation has made many achievements in improving separation performance and separation efficiency. However, whether these speech separation methods can be directly applied to bird sound separation remains to be studied for two reasons: (1) The human voice frequency is mainly within 4 kHz, and the sampling frequency usually is 8 kHz, while the bird sound frequency is higher, and the sampling frequency is usually 16 kHz or 32 kHz [30]. (2) There is a general dataset for speech separation, but no dataset for bird sound separation, and the speech separation dataset is not suitable for bird sound separation. Despite all this, the method used in speech separation does provide a method reference for the separation of mixed species bird sound. In this paper, we proposed an efficient bird sound separation network, dual-path tiny transformer network (DPTTNet), which can achieve good separation performance with less consumption of computing resources. Our main contributions can be summarized as follows:We built a mixed species bird sound dataset for bird sound separation.We first introduced a separation network with good separation performance to separate mixed species bird sound.We improved the separation efficiency of the separation network by using the simplified transformer structure.Two performance metrics and five efficiency metrics were used to compare the separation performance and efficiency of two speech separation networks and our proposed bird sound separation network on our self-built bird sound dataset.

## 2. Materials and Methods

### 2.1. Data

The dataset of our bird sounds separation was selected from the BirdCLEF2020 competition [31] and BirdCLEF2021 competition [32] datasets, with the data originally contributed by xeno-canto [33], one of the biggest birds sounds sharing website around the world. Some audio files with less noise and only one kind of bird sound left after clipping were selected to construct the original bird sound separation dataset, which contains 20 different species of bird sounds. These audio samples are in 16-bit wav format with a 32 kHz sampling frequency. The frequency ranges of all bird sounds and the spectrograms of some bird sound samples in the dataset are shown in Figure 2.

Noise refers to other background sounds that exist when recording bird sounds in the field, such as human sounds, insect sounds, wind, rain, and stream water sounds. In this paper, we use human sounds and insect sounds as noise. Additionally, all the noise data came from the sounds that were recorded in the northern, central, and southern urban forests of Guangzhou, including Shimen National Forest Park (SM), Maofeng Mountain Forest Park (MF), and Dafu Mountain Forest Park (DF), with a clear urban–rural gradient. The SM is located in an exurban area, the MF in a suburban area, and the DF in an urban area. All recording sites were set in typical southern subtropical evergreen broad-leaved forests, and dominant species include Machilus nanmu, Castanopsis fissa, Liquidambar formosana, and Acacia confusa. Three sound collection points were set up within each of the forest parks SM, MF, and DF, based on functional zoning, road distribution, and other human interference factors, for a total of nine sound collection points, ensuring that the sounds collected in this study were representative.

### 2.2. Data Preprocessing

The original audio data of the bird sounds separation dataset were of different lengths, which was not convenient for various subsequent processing, so the bird sound data needed to be clipped. All bird sound data with a duration of more than 24 s were cut into 24 s audio, and if the remaining duration was more than 8 s, it was saved as another audio file. The frequencies of all 20 bird species in the dataset are basically below 8 kHz, so the sampling frequency of 16 kHz is sufficient. Here, the sampling frequency of the audio is set to 16 kHz to reduce computational requirements and to preserve most of the frequency information of the audio. The training data can be generated by randomly selecting two different bird sounds and mixing them according to the following formula:(1)$$s\left(t\right)=s_{1}\left(t\right)+\alpha \cdot s_{2}\left(t\right)=s_{1}\left(t\right)+\sqrt{\frac{\sum_{t}{s_{1}}^{2}\left(t\right)}{10^{\frac{q}{10}}\sum_{t}{s_{2}}^{2}\left(t\right)}}\cdot s_{2}\left(t\right)$$
where *q* stands for the relative levels of $s_{1}\left(t\right)$ and $s_{2}\left(t\right)$, and its value is set to a random number in a range of −5 and +5 to ensure that the sound levels of the two bird sounds are different before mixing. Test data can also be generated based on this formula. In addition, the mixed data need to be normalized to prevent data overflow. The training dataset contains the sounds of 20 different species of birds, with a total of 51,615 mixed audio files of 148.32 h in duration. The test dataset contains the sounds of 20 different species of birds, with a total of 3789 mixed audio files of 10.93 h in duration. Additionally, the noise data were added to the mixed species bird sounds with a random SNR of 0–25 dB when studying the effect of noise on the separation performance. There are some samples of the mixed species bird sound dataset in the Figure 3.

### 2.3. Methodology

The mixed species bird sound signal can be expressed as a superposition of *C* bird sounds with different gains, expressed by the formula:(2)$$x\left(t\right)=\sum_{i=1}^{C}\alpha_{i}s_{i}\left(t\right)$$
where $x\left(t\right)\in R^{1\times T}$ is the mixed species bird sound signal, *T* denotes the length of signal, $\alpha_{i}$ denotes the gain of the ith bird sound, $s_{i}\left(t\right)\in R^{1\times T}$ denotes the ith source bird sound, and *C* is the number of mixed source bird sound; here, we take C=2, and we mainly study the separation under the mixture of two different kinds of bird sound. The purpose of single-channel mixed species bird sound separation is to separate two source bird sounds $s_{i}\left(t\right)$ from the mixed species bird sound x(t). ${\hat{s}}_{i}\left(t\right),i=1,\cdots ,C$ is the estimation of the bird sound separation network, and we need to make ${\hat{s}}_{i}\left(t\right)$ approach $s_{i}\left(t\right)$.

As shown in Figure 4, our proposed time-domain end-to-end bird sound separation network consists of three parts: encoder, separator, and decoder, and its structure is similar to DPTNet in the paper [23]. First, mixed audio is transformed into feature vectors using the encoder. Then, the separator uses the feature vectors as input to estimate the corresponding mask for each source. Finally, the mask and the feature vectors are then multiplied to obtain the estimated feature vector of each source bird sound, which is input to the decoder to reconstruct the source waveform. The encoder, separator, and decoder will be described in detail below.

#### 2.3.1. Encoder

The mixed species bird sound $x\left(t\right)\in R^{1\times T}$ is input to the encoder and transformed into feature vectors $w\in R^{N\times L}$, where *T* is the length of the input mixed species bird sound audio signal, *N* is the kernel size of a 1D convolutional layer in the encoder, and *L* is the length of the encoded features. The main component of the encoder is a 1D convolutional layer that simulates STFT to generate feature representations with the aim of learning more reasonable and appropriate mappings from the data through the network itself. As a rule of thumb, the convolutional stride is often set to half the kernel size to maintain continuity between features. The 1D convolutional layer is followed by a rectified linear unit (ReLU) activation function to ensure the non-negativity of the features, which can be summarized by the following equation:(3)w=ReLU(Conv1D(x))

#### 2.3.2. Separator

The core of this proposed separation network is the separator, whose structure is the same as the common dual-path network [21,23], which mainly consists of four parts: segmentation, DPTTNet block, dual-path block processing, and overlap-add, where the DPTTNet block plays an important role in reducing the computational load and parameters of the entire network.

1.Segmentation

The segmentation splits the feature w into *S* mixed chunks of length *K*, with 50% overlap between two adjacent chunks to maintain the association between different chunks, and then concatenates all the divided chunks into a 3D tensor $D\in R^{N\times K\times S}$ to facilitate the overall modeling of the input features later. If the length of the input features does not satisfy the segmentation condition, the input features need to be zero-padded in this study. A schematic diagram of the segmentation is shown in Figure 5.

2.DPTTNet block

The DPTTNet block is an improvement of transformer encoder and DPTNet block. Transformer encoder consists of three parts: scaled dot-product attention, multi-head attention, and feed-forward network. Scaled dot-product attention is a very efficient form of self-attention, where the weight for each position is computed using dot product and then normalized to obtain a weight vector. The output is the dot product of the weight vector and the input, as shown in Figure 6a. Multi-head attention consists of multiple scaled dot-product attentions, as shown in Figure 6b. First, the input features are linearly mapped multiple times through different linear layers to obtain the input features (queries, keys, and values) of the scaled dot-product attention. Then, the scaled dot-product attention is computed simultaneously on these mapped queries, keys and values, and the obtained results are concatenated together for a linear map to obtain the output of multi-head attention. The feed-forward network consists of two linear layers and a ReLU activation function. Each value in the bird sound audio sequence is not separate but is linked back and forth, but the position information in the bird sound audio sequence is not utilized by the transformer. The original transformer adds position encoding to the input to represent the position information, and the position encoding is generally performed with sine and cosine functions, or learnable parameters. The paper [23] found that position encoding is not suitable for dual-path networks and often leads to model scattering during training, so they replaced the first fully connected layer in the feed-forward network with an RNN to obtain an improved transformer structure, i.e., DPTNet block, whose structure is shown in Figure 7a. Although this improves the separation performance, it also loses the advantages of the original transformer parallel computing, because the RNN cannot perform parallel calculation. Moreover, the original transformer needs to use a lot of large matrix operations when computing the queries, keys, and values. The computational complexity of each feature computation is $O(L^{2})$, where *L* is the length of the input features, which makes the network computation requires large memory, a long training time, and high requirements for the computing equipment.

Considering these problems, we proposed a DPTTNet block, whose structure is shown in Figure 7b. We add a 1D convolutional layer in front of the original transformer to learn the position information, and the stride of the convolutional layer is 2. In this way, the length of the feature after convolution is halved, and the computational complexity becomes $O(\frac{L^{2}}{4})$ every time the query, keys, and values of the feature are calculated. In addition, due to the large amount of computation in computing long sequences, we remove the two fully connected layers in the feed-forward network to further reduce the amount of computation and parameters, and then add a deconvolution layer to restore the feature length to ensure the same length of input and output. The Algorithm 1 flow is as follows:
**Algorithm 1:** DPTTNet block forward.**Input**: $X\in R^{N\times K\times S}$**Output**: $Y\in R^{N\times K\times S}$Z=conv1d(X)**for***i = 1 to h***do**
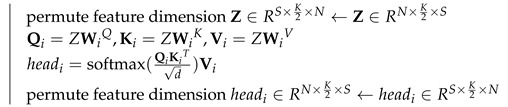
**end**($MultiHead=\left[\mathrm{concat}\left(head_{1},\ldots ,head_{h}\right)\right]W^{O}$Mid=LN(Z+MultiHead)Y=conv1d-transpose(max(0,Mid))

Where ${W_{i}}^{Q},{W_{i}}^{K},{W_{i}}^{V}\in R^{N\times \frac{N}{h}}$ represent the mapping parameter matrix of queries, keys, and values, respectively, $Q_{i},K_{i},V_{i}\in R^{\frac{K}{2}\times \frac{N}{h}}$ represent the mapped queries, keys, and values, respectively, $W^{O}$ is also a parameter matrix, *h* represents the number of heads in multi-head attention, and LN represents layer normalization [34].

3.Dual-path block processing

In the dual-path block processing stage, each complete sequence modeling process consists of two parts: intra-transformer and inter-transformer. As shown in Figure 4, the output **D** of the segmentation is passed to intra-transformers and inter-transformers for intra-block local modeling and inter-block global modeling: First, the intra-transformer acts on the second dimension of **D** to locally model each segment:(4)$$\begin{array}{cc} {D_{b}}^{\mathrm{intra}} & =\mathrm{IntraTransforme}r_{b}\left[{D_{b-1}}^{\mathrm{inter}}\right]\\ \; & =[\mathrm{transformer}\left({D_{b}}^{\mathrm{inter}}\left[:,:,i\right]\right),i=1,\ldots ,S] \end{array}$$

Then, the inter-transformer acts on the last dimension of *D*, collecting information from all blocks and learning global dependencies:(5)$$\begin{array}{cc} {D_{b}}^{\mathrm{inter}} & =\mathrm{IntraTransforme}r_{b}\left[{D_{b-1}}^{\mathrm{intra}}\right]\\ \; & =[\mathrm{transformer}\left({D_{b}}^{\mathrm{intra}}\left[:,j,:\right]\right),j=1,\ldots ,K] \end{array}$$
where *b* denotes the number of dual-path block processing, and when b=0, ${D_{0}}^{intra}$ denotes the input feature **D** after segmentation. It should be noted that the layer normalization of each intra-transformer and inter-transformer is applied to all dimensions.

The computational process of dual-path block processing can be expressed as:(6)$$D_{b+1}=f_{\mathrm{inter}}\left(\rho \left(f_{\mathrm{intra}}\left(D_{b}\right)\right)\right)$$
where $f_{inter}\left(\cdot \right)$ and $f_{intra}\left(\cdot \right)$ denote the inter-transformer and intra-transformer, respectively, and ρ denotes the exchange of the last two dimensions of $D_{b}\in R^{N\times K\times S}$. $D_{B}$ is obtained after *B* dual-path block processing, and then a one-dimensional convolution is taken to turn the output channel into C∗N, where *C* is the number of sound sources.
(7)$$D_{\mathrm{output}}=\psi^{-1}\left(f_{\mathrm{output}}\left(\psi \left(D_{B}\right)\right)\right)$$
where $f_{output}\left(\cdot \right)$ denotes one-dimensional convolution, and ψ denotes the reshape operation on the input features. Because a 1D convolutional layer cannot directly operate on $D_{b}\in R^{N\times K\times S}$, it is necessary to combine the last two dimensions into one dimension, and then to inverse transform back after convolution.

4.Overlap-Add

The $D_{output}$ obtained after the dual-path block processing needs to be converted into the same shape as before the segmentation by mixed and adding. The computation process is the reverse operation of the previous segmentation, and then the estimated masks of C sound sources are obtained after the 1D convolution and activation function.
(8)$$m_{i}=\mathrm{ReLU}\left(\mathrm{Conv}1D\left(\mathrm{OverlapAdd}\left(D_{output}\right)\right)\right)$$
where OverlapAdd represents the overlap and addition of $D_{output}$. When added, each segment of the feature has a 50% overlap.

#### 2.3.3. Decoder

The decoder, with the same kernel size and stride as the encoder, mainly consists of a transposed convolutional layer, which reconstructs the high-dimensional feature vector into the bird sound signal. Using the estimated mask $m_{i}\in R^{N\times L}$ of different sound sources obtained in the separator and the output $w\in R^{N\times L}$ of the encoder, the dot product of $m_{i}$ and *w* is transformed in the decoder to obtain the separated target bird sound, and the transformation can be expressed as:(9)$${\hat{s}}_{i}\left(t\right)=\mathrm{transpose}\_\mathrm{Conv1D}\left(w⊙m_{i}\right)$$
where ⊙ represents element-wise multiplication, and ${\hat{s}}_{i}\left(t\right)\in R^{1\times T}$ represents the estimate of $s_{i}\left(t\right)$.

### 2.4. Evaluation Methods

Source to Distortion Ratio improvement (SDRi) [35] and Scale-Invariant Source-to-Noise Ratio improvement (SI-SNRi) [15] are important indicators for evaluating the separation performance, and the larger the value obtained from the calculation, the better the separation performance. SDRi can be calculated using the calculation method in the Python library mir_eval [36]. Additionally, SI-SNRi is defined as:(10)$$\mathrm{SI}-\mathrm{SNR}=10\log_{10}\frac{{∥s_{target}∥}^{2}}{{∥e_{noise}∥}^{2}}$$
(11)$$s_{target}=\frac{\langle \hat{s},s\rangle s}{{∥s∥}^{2}}$$
(12)$$e_{noise}=\hat{s}-s_{target}$$
(13)$$\mathrm{SI}-\mathrm{SNRi}=\mathrm{SI}-\mathrm{SNR}\left(\hat{s},s\right)-\mathrm{SI}-\mathrm{SNR}\left(\hat{s},x\right)$$
where $\hat{s}$, s represent the estimated and label bird sounds, respectively, and **x** is the mixed species bird sound.

In order to better evaluate the computational efficiency and the occupation of computational resources of the separation network, we use the following evaluation indicators:Number of executed floating point operations (FLOPs);Number of trainable parameters;Memory allocation required on the device for a single pass;Time for completing each process.

An efficient separation network usually has lower FLOPs, parameters, and usage of memory and runtime on the premise of ensuring separation performance, which enables us to better train the separation network on devices with limited computational resources, and we can easily deploy the network in mobile devices and edge computing devices.

### 2.5. Experimental Setup

In this paper, the training and testing environment of the bird sound separation network is shown in Table 1.

In the experiment, the convolution kernel size enc_kernel_size is set to 16, the stride is set to 8, and the number of convolution kernels of the encoder convolutional layer and the decoder deconvolutional layer in Figure 4 is set to 256. In the separator, the length of segmentation *K* is set to 120, the number of dual-path transformer processing *B* is set to 6, and the number of heads of multi-head attention in each dual-path tiny transformer block is set to 4. During training, a batch of audio data is randomly selected, and the data are randomly clipped into 4s audio. The input dimension is represented as batch × C × T = batch × 1 × 64,000, where *C* represents the number of channels, and *T* represents the length of the data. Adam [37] is used as the optimizer. Gradient clipping with a maximum $L_{2}$-norm of 5 is applied for all experiments. Each network is trained for 100 epochs and stop with the criterion that the loss function do not degenerate on the validation set for 10 epochs. The warm-up is performed in the first *n* training steps to linearly increase the learning rate, and then the learning rate decays by 0.98 every two epochs:(14)$$lr=\left\{\begin{array}{c} k_{1}\times d_{model}^{-0.5}\times n\times warmup\_n^{-1.5},n\leq warmup\_n\\ k_{2}\times 0.98^{epoch/2},n>warmup\_n \end{array}\right.$$
where *n* is the number of steps, $k_{1}$ and $k_{2}$ are adjustable scalars, $k_{1}=0.2$, $k_{2}=1.5\times 10^{-4}$, and warmup_n=4000 is set in this paper. The settings of these hyper-parameters refer to DPTNet and DPRNN. Furthermore, to make the experimental results more convincing, the parameter settings of DPRNN and DPTNet used in this paper are the same as the best experimental parameter settings in the paper [21,23], except for the encoder parameter settings. The training method is also the same as the literature [21,23], and the relevant codes for DPRNN and DPTNet training are all from the author’s public code. When training the separation network with noise, the input data are a random mix of noise and mixed species bird sounds with a SNR ranging from 0 to 25 dB, and the labeled data are clean bird sound signals.

One thing that needs to be explained is that we do not use the settings of enc_kernel_size=2 and K=250 because doing so will greatly increase FLOPs, memory occupation, and training time, which is contrary to the original intention of this study, although choosing that setting will have better SI-SNRi in the experiments.

SI-SNR was used as the loss function during training, as it was commonly used as an evaluation metric for source separation, and the proposed network was trained using utterance-level permutation invariant training (uPIT) [38] to maximize SI-SNR.

## 3. Results

### 3.1. Separation Performance Analysis

Table 2 shows the comparison of the separation performance and network parameters of our proposed bird sound separation network with other speech separation networks on the mixed species bird sound dataset. As can be seen, our proposed bird sound separation network achieves excellent separation performance (19.3 dB for SI-SNRi and 20.1 dB for SDRi). Compared with DPTNet, although the separation performance of our network is reduced by 2.2 dB and 2 dB in SI-SNRi and SDRi, respectively, the number of parameters is greatly reduced: the number of parameters is only 0.4 M, which is 16.8% of DPTNet. Compared with DPRNN, our proposed bird sound separation network achieves the same separation performance, while the number of parameters is reduced by 83%.

The separation results of this proposed bird sound separation network can also be illustrated by the spectrogram in Figure 8. As can be seen, the spectrograms (Figure 8b,c) separated from Figure 8a are very similar to the spectrograms of original clean bird sounds (Figure 8g,h). Figure 9 shows some other samples in the test dataset, and we can see that the mixed species bird sounds are well separated. The training method in the paper [38] is used to enhance the separation performance of this network in separating mixed species bird sounds in noisy environments. As shown in Figure 10, when mixed species bird sounds with different signal-to-noise ratios are input during verification, the bird sounds separated by the network that trained with noise have better SI-SNR, which proves that our network can be more robust through the use of the special training method. Additionally, as shown in Figure 8, the mixed species bird sounds with noise Figure 8d are separated by this network trained in this method to obtain clean bird sounds (Figure 8e,f), which indicates that our separation network also has noise reduction capabilities.

### 3.2. Computational Resource Analysis

In order to analyze the effect of the sampling frequency of the audio on the FLOPs and runtime of different separation networks, a comparison experiment was carried out to show their relationship. From the experimental results (Table 3), we can see that the FLOPs and runtime of the separation network almost doubled whenever the sampling frequency doubled. The FLOPs and runtime of our proposed bird sound separation network are smaller than those of the other two speech separation networks at each sampling frequency. The following data are obtained from the audio with a duration of 4 s.

Table 4 shows the runtime and memory occupation for different networks, where CPU Time denotes the time spent when the network running in the CPU without using the GPU, and GPU Time denotes the time spent when the network running in the GPU; F represents network forward and B denotes network backward. For forward inference in the CPU, our separation network only requires 0.316 s, while DPRNN and DPTNet require 2.047 s and 2.256 s, respectively, and for running in the GPU, our separation network only requires 13.507 ms, while DPRNN and DPTNet require 54.994 ms and 52.001 ms, respectively. In memory occupation, our separation network is 200 MB and 300 MB less than DPRNN and DPTNet on forward and backward, respectively, and nearly 1000MB less than DPTNet on backward.

Furthermore, the effect of the convolution kernel size TCK and convolution stride TCS of the convolutional layers in the tiny transformer module on the separation performance is studied. As can be seen from Table 5, with a gradual decrease in TCK and TCS, the separation performance of the network gradually improves, but the runtime increases.

If the training FLOPs is limited, different separation networks also have different performances. From the view of the separation performance result (Figure 11), our separation network can achieve good performance with fewer FLOPs in training, and the separation performance outperform DPTNet in the validation separation of 500 1000 Peta. Compared with DPRNN, our separation network can achieve better validation SI-SNRi under the same training FLOPs.

## 4. Discussion

Speech separation has made great progress in recent years with the support of deep learning, but very little work has been conducted on the separation of mixed species bird sounds. In addition, there was a big problem in bird sound separation; there was no relevant dataset. The currently public bird sound datasets were mainly used for identification tasks, and the audio in the dataset usually contained multiple bird sounds. Even though these audios were given separate labels, they were not suitable for bird sound separation. In this study, we evaluated the performance of the bird sound separation network in a self-built bird sound dataset, and speech separation networks were applied to the separation of bird sounds. As we can see, the speech separation network does have a good effect for mixed species bird sound (DPRNN’s SI-SNRi reaches 19.3 dB and SDRi reaches 20.0 dB in the mixed species bird sound dataset; DPTNet’s SI-SNRi reaches 21.5 dB and SDRi reaches 22.1 dB in the mixed species bird sound dataset), but there are still some problems. A significant difference between bird sound separation and speech separation is that the sampling frequency of bird sound is usually higher. For example, in speech separation, the sampling frequency of the audio is usually only 8 kHz, while in bird sound separation, a higher sampling frequency is needed because the frequency of bird sound is higher, and the sampling frequency of bird sound signals is usually 16 kHz or 32 kHz. For a fixed period of bird sound, a higher sampling frequency means more data. The time-domain single-channel end-to-end source separation network is quite sensitive to changes in data length (Table 3), and the FLOPs and runtime required by the separation network to separate a segment of sound doubles every time when the sampling frequency doubles. If the speech separation network requires high computational resources for separating low sampling frequency signals, this increase in computational resources will become unacceptable when the sampling frequency of the audio doubles or triples. Therefore, to separate the mixed species bird sounds needs a more efficient separation network.

Our proposed bird sound separation network is a dual-path network similar to DPRNN and DPTNet, the difference of the network is only the structure used to extract features. DPRNN uses LSTM, DPTNet uses DPTNet block (an improved structure of transformer, Figure 7a), while our network uses DPTTNet block (a simplified transformer structure, Figure 7b) to extract features. The experimental results show that our bird sound separation network has excellent separation performance in the mixed species bird sound dataset (SI-SNRi reaches 19.3 dB and SDRi reaches 20.1 dB), and it is better than DPRNN and DPTNet in FLOPs, runtime, memory occupation, and parameters. Furthermore, our network has a greater advantage over DPRNN and DPTNet for signals with 8 kHz, 16 kHz, and 32 kHz sampling frequency. The FLOPs and runtime of our network are fewer in the case of 32 kHz sampling frequency audio than those of DPRNN and DPTNet in the case of 8 kHz sampling frequency audio. In addition, the parameters of our network are only 0.4 M, which means that it needs only a small space to store the network parameters. These advantages enable our network to train faster, and can be easily deployed and applied in various mobile and edge computing devices with limited computing resources, which is very important for on-site bird monitoring. To further investigate the acceleration performance of our proposed network, we also investigated the convolutional kernel size and convolutional stride in the DPTTNet block (Table 5). With the gradual increase in the size of the convolutional kernel and the convolutional stride, the separation performance of the network is gradually reduced, but the runtime is also gradually reduced, which means a faster separation speed. Good separation performance is not required in all cases, and sometimes a faster separation speed is preferred under certain separation performances. Therefore, in practice, the network with appropriate convolutional kernel size and convolutional stride can be selected according to the desired separation performance to achieve a balance between separation performance and speed.

Although a lot of work has been conducted on the performance and efficiency of bird sound separation, there is still room for improvement. In this paper, only two kinds of bird sounds are separated, but in reality, the number of kinds of mixed species bird sounds is unknown, so it is necessary to judge the number of bird sounds in mixed sounds and separate them. What’s more, embedding transformers in the dual-path network can improve the separation performance, but at the cost of more training data to train the network. However, the target bird sound to be separated does not always have enough corresponding training data, so a good pre-training model is required, and the pre-training model can be used to quickly fine-tune the desired effect in small datasets. Furthermore, the amplitude of bird sound after separation is different from that before mixing, which requires further optimization of the separation network and our further research. Noise is also a major problem of bird sound separation, and we have tried to add noise to the training to combat noise interference (Figure 10), but the effect is limited. If the noise can be minimized at the same time of separation, the bird sound separation network will be more robust. We can also improve separation efficiency by using some newly proposed more efficient network structures, and by modeling compression methods.

## 5. Conclusions

Mixed species bird sounds can make it difficult to perform subsequent related tasks such as the identification of bird sounds or the estimation of species richness according to acoustic indices, so it is necessary to separate mixed species bird sounds. The current time-domain single-channel end-to-end source separation network requires unacceptable computational resources for separating bird sounds with high sampling frequency, and we need a more efficient bird sound separation network. In this paper, we proposed an efficient time-domain single-channel end-to-end bird sound separation network, which requires less computational resources while achieving excellent separation performance and fast separation speed. The proposed network was evaluated on separation performance and efficiency, and our work may contribute to discriminate individual birds and to study the interaction between individual birds, as well as to realize the automatic identification of bird species in various mobile and edge computing devices.

## Figures and Tables

**Figure 1 animals-12-03117-f001:**
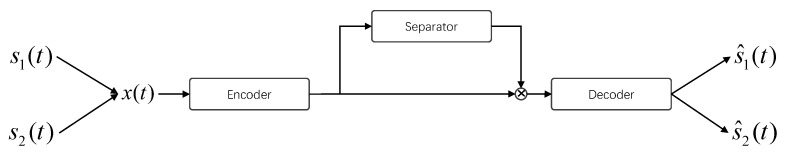
Simple representation of bird sound separation structure based on deep learning.

**Figure 2 animals-12-03117-f002:**
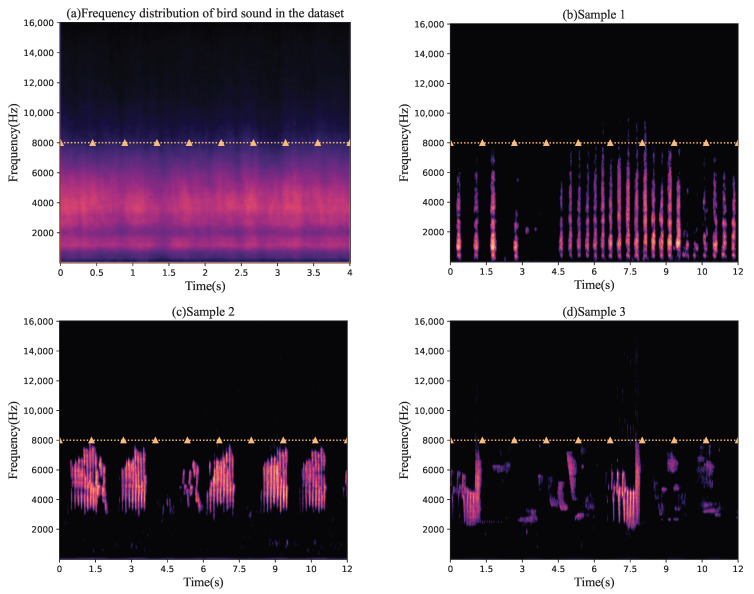
Frequency distribution of bird sounds in the dataset and some samples.

**Figure 3 animals-12-03117-f003:**
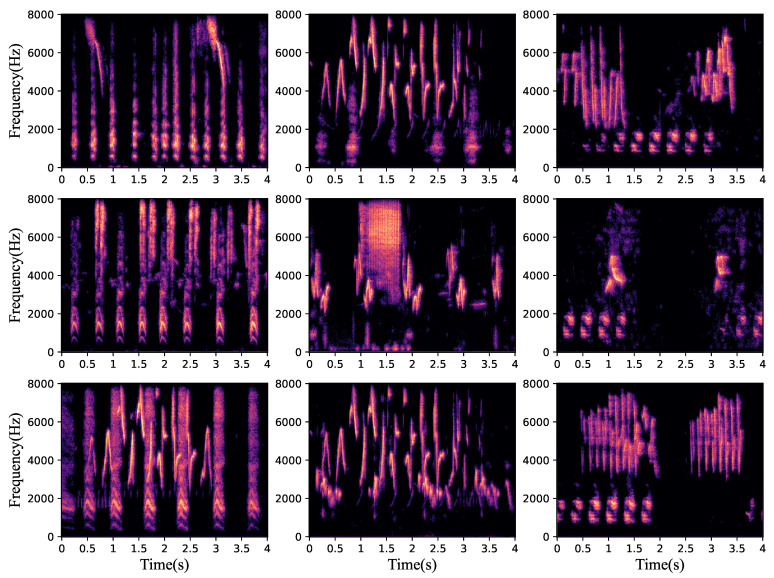
Some samples in the mixed species bird sound dataset.

**Figure 4 animals-12-03117-f004:**
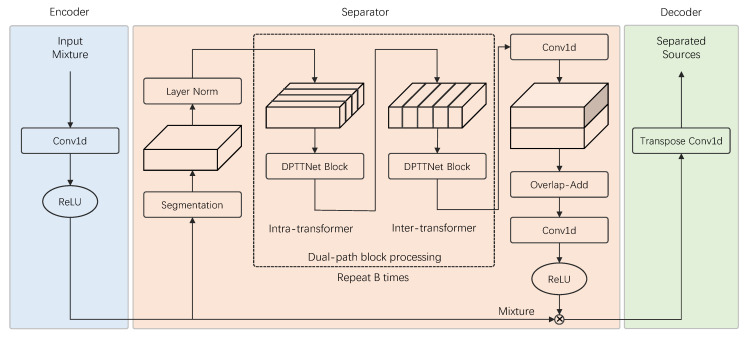
Framework of bird sound separation with dual-path tiny transformer network.

**Figure 5 animals-12-03117-f005:**
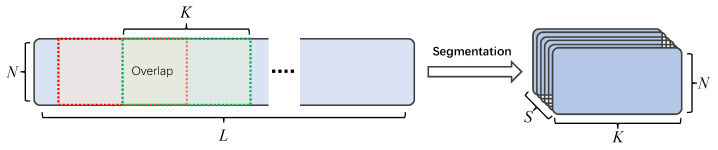
Feature segmentation diagram.

**Figure 6 animals-12-03117-f006:**
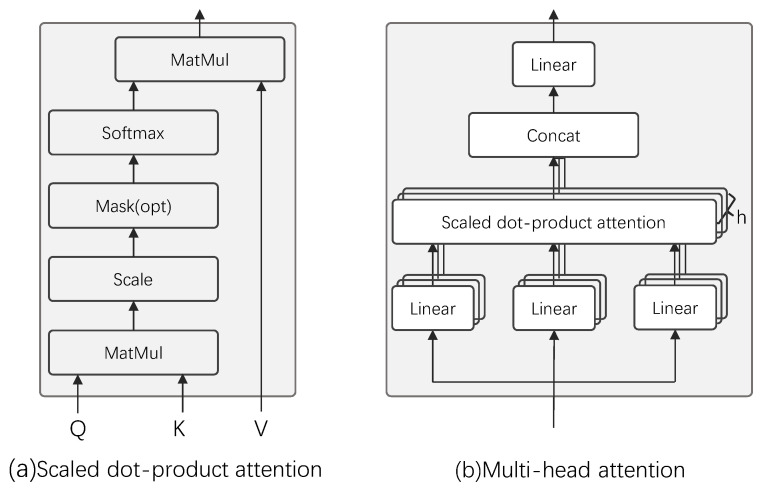
Attention mechanism in transformer.

**Figure 7 animals-12-03117-f007:**
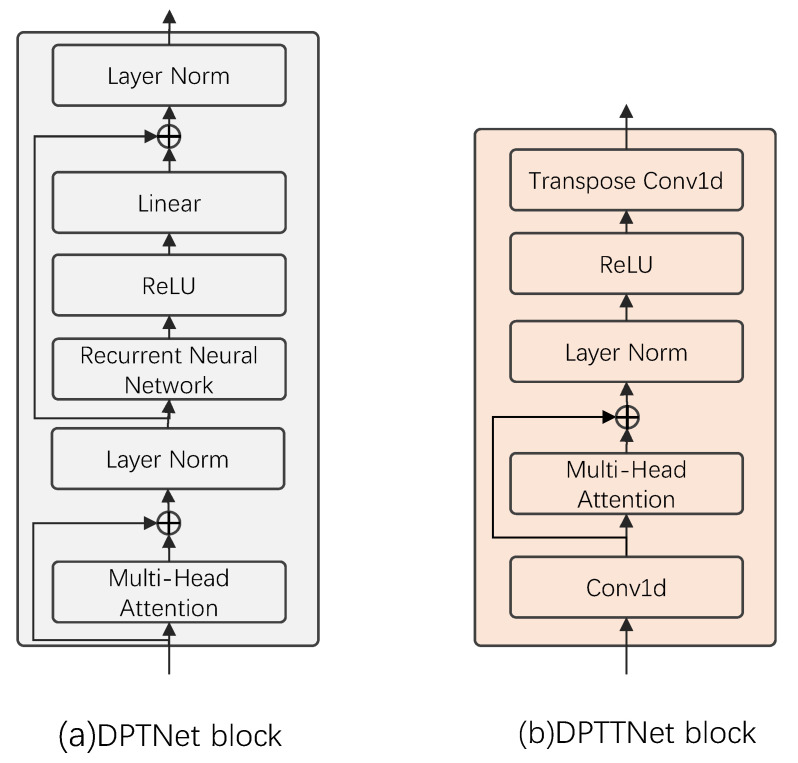
DPTNet block and DPTTNet block.

**Figure 8 animals-12-03117-f008:**
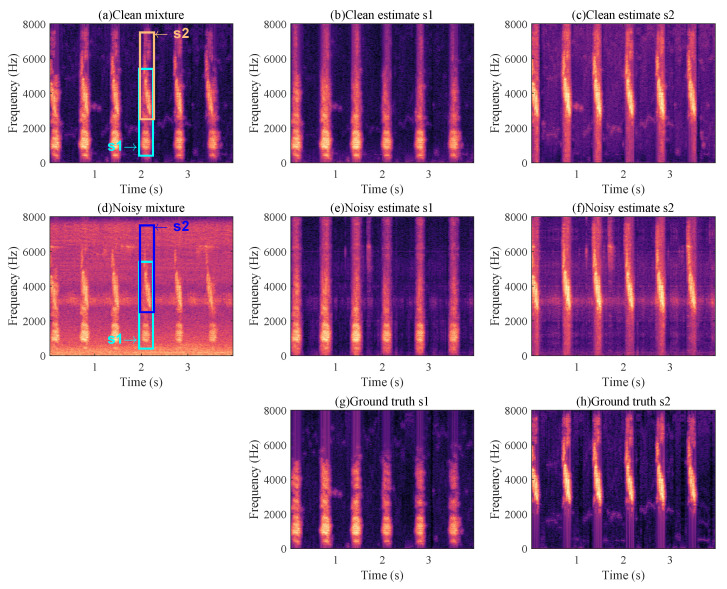
Separation results. (**a**) represents the spectrogram of mixed species bird sound, (**b**,**c**) represents the bird sound separated from (**a**). (**d**) represents the mixed species bird sound after mixing the noise, and (**e**,**f**) represents the bird sound separated from (**d**). (**g**,**h**) are clean original bird sounds.

**Figure 9 animals-12-03117-f009:**
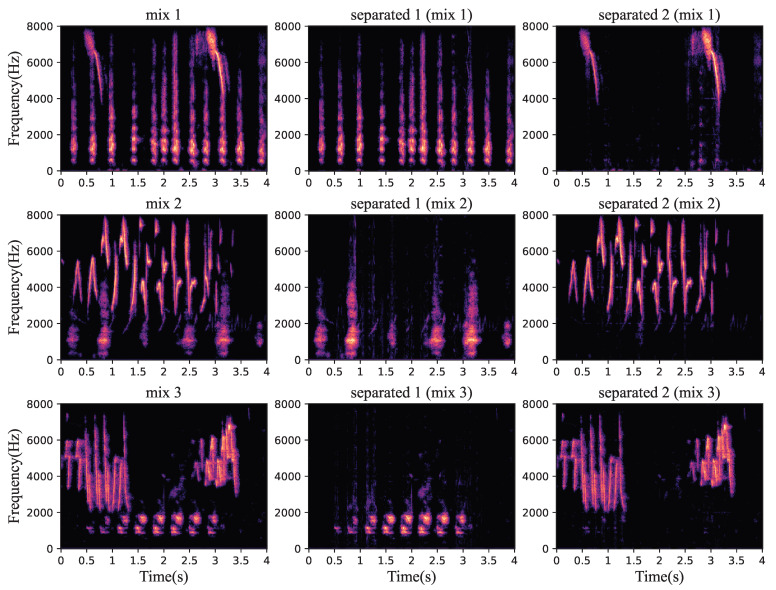
Some separation results of the dataset.

**Figure 10 animals-12-03117-f010:**
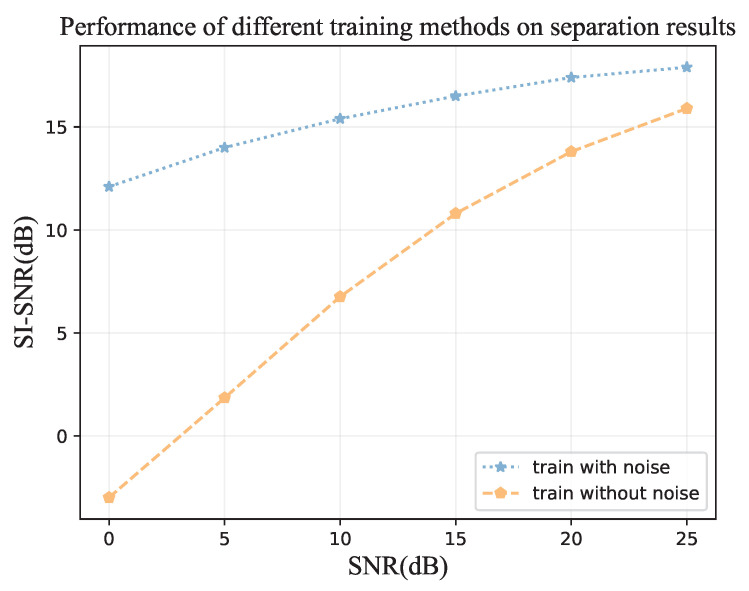
In the case of training with noise and training without noise, different SI-SNRs were obtained when inputting audio with different SNRs during validation. When training with noise, the input audio mixes the noise and the mixed species bird sound with a random SNR of 0–25 dB.

**Figure 11 animals-12-03117-f011:**
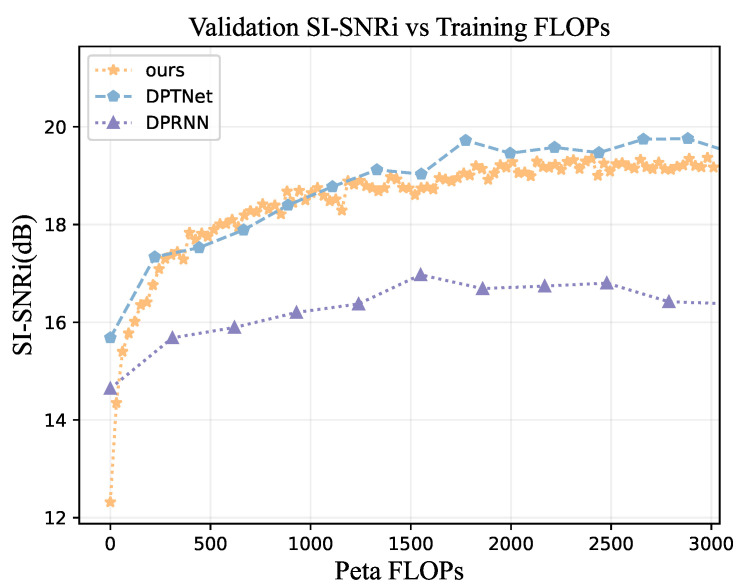
The separation performance obtained when different FLOPs are used for training. The horizontal axis indicates the FLOPs used to train the network, and the vertical axis represents the SI-SNRi obtained from the validation of the network trained under these FLOPs.

**Table 1 animals-12-03117-t001:** Training and testing environment.

Environment	Setup (Training)	Setup (Testing)
GPU	NVIDIA GeForce RTX 3080 Ti 12 G	NVIDIA GeForce RTX 3060 12 G
CPU	Intel(R) Core(TM) i7-11700F	Intel(R) Core(TM) i5-12400F
RAM	32 G	16 G
Operating system	Windows 10 64 bit	Windows 10 64 bit
Software environment	Python 3.8.10, Pytorch 1.11, CUDA 11.6	Python 3.8.10, Pytorch 1.11, CUDA 11.6

**Table 2 animals-12-03117-t002:** Separation performance of different network on the bird sounds dataset.

Network	SI-SNRi (dB)	SDRi (dB)	Params (M)
DPRNN	19.3	20.0	2.6
DPTNet	21.5	22.1	2.6
Ours	19.3	20.1	0.4

**Table 3 animals-12-03117-t003:** Floating point operations and runtimes of different networks for inputting bird sounds with different sampling frequencies.

Network		FLOPs (G)			CPU Time (s)			GPU Time (ms)	
	**8 kHz**	**16 kHz**	**32 kHz**	**8 kHz**	**16 kHz**	**32 kHz**	**8 kHz**	**16 kHz**	**32 kHz**
DPRNN	30.374	60.016	119.302	1.066	2.047	3.598	31.994	54.994	109.167
DPTNet	21.742	42.961	85.402	1.106	2.256	4.407	29.839	52.001	107.494
Ours	2.943	5.893	11.651	0.141	0.316	0.673	8.167	13.507	30.506

**Table 4 animals-12-03117-t004:** Floating point operations, runtimes, and memory occupations of different networks. The best results are shown in bold font.

Model	FLOPs (G)	CPU Time (s)	GPU Time (ms)	F/B GPU Memory (GB)
DPRNN	60.016	2.047	54.994	0.890/1390
DPTNet	42.961	2.256	52.001	0.844/1.996
Ours	5.893	0.316	13.507	0.602/1.002

**Table 5 animals-12-03117-t005:** Time spent and separation performance of our network with different convolutional kernel sizes and convolutional strides in DPTTNet block.

TCK	TCS	CPU Time (s)	SI-SNRi (dB)	SDRi (dB)
16	8	0.185	16.6	17.5
8	4	0.245	18.8	19.6
4	2	0.316	19.3	20.1

## Data Availability

Bird sound data and other related data can be obtained from https://www.kaggle.com/datasets/chenyonghuan/bird-separation-dataset, accessed on 9 August 2022.

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
