# Peer review of "An Efficient Time-Domain End-to-End Single-Channel Bird Sound Separation Network"

_animals, 2022, doi:10.3390/ani12223117_

Round 1

Reviewer 1 Report

This is an interesting and useful study, providing solutions for bird species identifications using bird sounds. The network could generate relatively accurate separations and had higher efficiency. The paper is well written and I only have some minor suggestions below.

1.      Although the English writing is of high quality, it can be improved further. For example, ‘bird sounds’ occurred twice on lines 2-3; passive voice is suggested for your work.

2.      L4, ‘classification’ to ‘identification’

3.      L24, ‘classify’ to ‘identify’

4.      L158, I suggest not describing the organization of the paper, but highlighting the potential implications of the network.

5.      L248, you only used one audio file in the experiment. I think the sample size was too small. I suggest testing the network for more samples.

Reviewer 2 Report

It is clear that these authors have executed some complex scientific analysis of a large dataset. And this is commendable. The issue with this article is that it is not written up in the correct style of a research article and does not follow the format or organisation of sections for the selected journal. The format and readability of the manuscript needs work. 

Does the paper not require a simple summary and an abstract?

There should be a separate results section and a discussion.

The explanation of the findings and the evaluation of the methodology in the currently conjoined "Results and Discussion" is not detailed enough to provide a thorough review of the findings and their wider importance. Separating these sections out would allow for a more considered review of research extensions that are based on the current study and findings.

Please provide links or further explanation of where the publicly available  birds sounds database is held. 

Section 3.1 (Experiments) reads more like methods.

The whole paper needs to be re-written and re-viewed so that it has an introduction that sets the scene, which ends with clear and predictions written out for the reader. Then the methods, that describe the dataset, the collection of results and the statistical testing. Then the results, a separate discussion and finally the conclusion. 

At the moment, the many different subheadings used that do not fit into the normal, logical way of writing a research article means it is very hard to follow.

I cannot evaluate or judge the relevance of the conclusions without being able to follow the methods fully or the explanation of the results section. 

The introduction is very long, at the expense of explanation of the results in the discussion. 

Figures and illustrations are useful but these need more description and context. A lot of technical background knowledge is expected of the reader and I recommend further description and explanation of the analyses, the equations stated and the graphical outputs. 

I think this paper is probably, eventually, worthy of publication, but at the moment, I am confused about what has been tested and why. What the overall aim of the research question is, and do not feel confident in the authors' evaluation and discussion of their results. 

Reviewer 3 Report

The paper is well organized, it gives a sufficent contribution.

Please make a revision of some paragraphs, the english could be improved.

All equations are whole part of the text and must also respect the punctuation rules, please include commas and dots. For example, in equation (1) include a comma at the final.

Round 2

Reviewer 2 Report

Useful revisions that are helpful to clarifying the understanding of the paper, its aims and scope. The summary of contributions of this paper to the literature is a useful one.

A few minor comments.

The simple summary is still very technical. For example "deep learning based automatic bird sound recognition" is not something that a non-scientist (or even a scientist that does not specialise in auditory research) will understand. The simple summary I

Throughout the paper, I recommend "mixed species bird sound" rather than "mixed bird sound". As the latter sounds like you are scrambling the sounds made by a single individual bird. Please edit wherever you include "mixed bird sound". 

In the discussion, line 379 to 380, I am unsure what is meant here. What resources? What is the area for improvement?

In the introduction, please define what is meant by noise. This is a human created term, so what sounds are defined as noise?
